# The Ocean Model for E3SM Global Applications: Omega Version 0.1.0. A New High-Performance Computing Code for Exascale Architectures

Mark R. Petersen<sup>1</sup>, Xylar S. Asay-Davis<sup>1</sup>, Alice M. Barthel<sup>1</sup>, Carolyn Branecky Begeman<sup>1</sup>, Siddhartha Bishnu<sup>2</sup>, Steven R. Brus<sup>3</sup>, Philip W. Jones<sup>1</sup>, Hyun-Gyu Kang<sup>4</sup>, Youngsung Kim<sup>4</sup>, Azamat Mametjanov<sup>3</sup>, Brian J. O'Neill<sup>1</sup>, Kieran K. Ringel<sup>1,5</sup>, Katherine M. Smith<sup>1</sup>, Sarat Sreepathi<sup>4</sup>, Luke P. Van Roekel<sup>1</sup>, and Maciej Waruszewski<sup>6</sup>

**Correspondence:** Mark R. Petersen (mpetersen@lanl.gov)

Abstract. Here we introduce Omega, the Ocean Model for E3SM Global Applications. Omega is a new ocean model designed to run efficiently on high performance computing (HPC) platforms, including exascale heterogeneous architectures with accelerators, such as Graphics Processing Units (GPUs). Omega is written in C++ and uses the Kokkos performance portability library. These were chosen because they are well-supported, and will help future-proof Omega for upcoming HPC architectures. Omega will eventually replace the Model for Prediction Across Scales-Ocean (MPAS-Ocean) in the US Department of Energy's Energy Exascale Earth System Model (E3SM). Omega runs on unstructured horizontal meshes with variable-resolution capability and implements the same horizontal discretization as MPAS-Ocean. In this paper, we document the design and performance of Omega Version 0.1.0 (Omega-V0), which solves the shallow water equations with passive tracers and is the first step towards the full primitive equation ocean model. On Central Processing Units (CPUs), Omega-V0 is 1.4 times faster than MPAS-Ocean with the same configuration. Omega-V0 is more efficient on GPUs than CPUs on a per-watt basis—by a factor of 5.3 on Frontier and 3.6 on Aurora, two of the world's fastest exascale computers.

### 1 Introduction

Ocean models have always required the fastest computers available to resolve the fine spatial scales and simulate the long timescales inherent in ocean circulation. The E3SM project in particular is focused on using the fastest computers fielded by the US Department of Energy (DOE) to perform high-resolution simulations of the Earth system, including the global ocean. Ocean modelers have always needed to adapt to underlying HPC architectures and changing programming models. The earliest global ocean models were written in Fortran in the 1960s (Bryan and Cox, 1968) and were later optimized in the 1970s and 1980s for vector supercomputers (e.g., Semtner and Chervin, 1988) to perform early eddy-permitting simulations. During the

<sup>&</sup>lt;sup>1</sup>Los Alamos National Laboratory, Los Alamos, NM 87545, USA

<sup>&</sup>lt;sup>2</sup>Department of Earth Sciences, University of Cambridge

<sup>&</sup>lt;sup>3</sup>Argonne National Laboratory, Lemont, IL 60439, USA

<sup>&</sup>lt;sup>4</sup>Oak Ridge National Laboratory, Oak Ridge, TN 37830, USA

<sup>&</sup>lt;sup>5</sup>Center for Nonlinear Studies, Los Alamos National Laboratory, NM, 87545, USA

<sup>&</sup>lt;sup>6</sup>Sandia National Laboratories, Albuquerque, NM 87123, USA

50

parallel computing transition in the late 1980s and 1990s, the Parallel Ocean Program (POP; Dukowicz et al., 1993) led the way with a data-parallel version of the Bryan-Cox models and the introduction of some algorithmic improvements needed for parallel implementations (Dukowicz and Smith, 1994). POP was used to perform the first eddy-resolving simulations of the global ocean and the North Atlantic (Maltrud and McClean, 2005; Smith et al., 2000). After nearly a decade of competition among parallel programming models in the 1990s, the Message Passing Interface (MPI, 2025,1993) eventually became the de facto standard programming model, and models like POP were adapted to the MPI paradigm with horizontal domain decomposition surrounded by halo points to reduce the number of messages between domains (e.g., Smith et al., 2010). As commodity microprocessors in HPC clusters began to include multiple Central Processing Unit (CPU) cores, OpenMP (2024) directives were added to MPI models for on-node thread-based parallelism among the cores (Wallcraft, 2000; Kerbyson and Jones, 2005).

Another transition in computing is now in progress, due to the power and cooling constraints that limit the performance of CPUs. The latest HPC architectures are heterogeneous systems with a number of different processing elements, most commonly featuring the pairing of CPUs with Graphics Processing Units (GPUs) as accelerators. In the current TOP500 ratings, only two of the top 50 machines are not equipped with GPUs (Strohmaier et al., 2025). Moreover, the majority of computing power in the largest machines resides in their GPUs. For example, Frontier, ranked second in the TOP500, provides 2.5 TFLOPs from the CPU on each node and 192 TFLOPs from its 4 GPUs (98.7%). Similarly, Aurora provides 27 TFLOPs from CPUs and 314 TFLOPs per node from GPUs (92%). Thus, to fully leverage the capabilities of modern HPC systems and achieve high performance, any computational physics model must be developed to run efficiently on GPUs.

The new Ocean Model for E3SM Global Applications (Omega) is no exception. Since our goal is for Omega to operate effectively on these heterogeneous architectures, it must be explicitly designed to harness the computational power of GPUs. As in the early 1990s parallel transition there are a number of competing programming models for these new heterogeneous architectures and few options that are portable across even the Leadership Computing Facilities of the DOE that deploy the fastest computing platforms for scientific computing. Currently, programmers have several choices: (1) adding directives like OpenACC (OpenACC, 2022) pragmas for offloading computation to the GPUs; (2) writing low-level GPU code using vendor-specific API (Application Programming Interface) like CUDA (NVIDIA Corporation, 2023), HIP (Advanced Micro Devices, Inc., 2023), or DPC++ (John et al., 2021); (3) using climate or weather domain-specific language or source-to-source compilers like PSyclone (2019) or GridTools (2019); (4) adopting performance-portable programming models such as Kokkos (Trott et al., 2022a), YAKL (Norman et al., 2022) and Raja (Beckingsale et al., 2019), providing high-level programmer-friendly abstractions, that are compiled down to optimized, vendor-specific backends. We define performance portability as the ability of code to achieve high performance across different computing platforms without requiring code modifications or performance tuning for each specific platform.

The vast majority of climate model components, including ocean models, are written in Fortran using MPI to share data between nodes and OpenMP for multi-threading within shared-memory nodes. The easiest route to running on GPUs is to add OpenACC directives around loops (option 1 above), as the base code remains unchanged and the format is similar to OpenMP directives. This approach was adopted by our group for MPAS-Ocean, and also by ICON-O (Porter and Heimbach, 2025).

60

75

MPAS-Ocean with OpenACC was successfully deployed on GPUs. However, due to the MPAS-Ocean structure, only half of the code could be accelerated with directives, resulting in limited performance gains. This ranged from modest speed-ups on some machines, such as Frontier, to slow-downs on others, such as Perlmutter, as described in Section 5. In all cases, performance with OpenACC was far below the expected throughput for each GPU node due to the small kernel sizes and inability to accelerate half the code. In addition, compiler support for Fortran was poor and delayed on many GPU architectures. These issues motivated us to look for a different GPU programming model and to pursue a complete re-write of the ocean model itself.

For option 2, each compute vendor has an API to program their devices, which is often proprietary. This results in vendor lock-in and lack of portability. These APIs are at a much lower level in the system software stack than the directives-based approaches described earlier. Nvidia provides the CUDA (Compute Unified Device Architecture) toolkit to program their GPUs and offload computation. With the right expertise, this can result in highly optimized code and excellent performance on Nvidia GPUs. AMD in turn introduced Heterogeneous-computing Interface for Portability (HIP), a C++ runtime API and kernel language that facilitates developing applications primarily for AMD GPUs. Although HIP promises NVIDIA GPU support, performance portability is not guaranteed. SYCL is an open C++ standard that adds data parallelism and heterogeneous programming to standard C++. Especially Intel GPUs can be programmed using Data Parallel C++ (DPC++), which is Intel's implementation of SYCL. However, these APIs are not beginner-friendly and have a steep learning curve for domain scientists. Basic portability itself is not guaranteed. Moreover, performance portability across diverse GPU architectures is not a design target and would require extensive optimization effort. Models that have had a version converted to run on GPUs with CUDA include the Princeton Ocean Model (Xu et al., 2014, 2015), the Finite Volume Coastal Ocean Model (FVCOM) (Zhao et al., 2017), and the Weather Research and Forecasting (WRF) (Mielikainen et al., 2012). This approach was not considered for our model due to portability concerns across the diverse set of DOE computing platforms.

Domain-specific languages (option 3) that are suitable for ocean models are limited and do not yet have large community buy-in, making them a risky choice for long-term development. Julia is a notable new option that supports performance portability across new architectures. The Climate Modeling Alliance (CliMA) has embraced Julia as its core language, and their ocean component, Oceananigans.jl (Ramadhan et al., 2020), has demonstrated strong computational performance on GPUs (Silvestri et al., 2025). While Julia continues to evolve and gain traction in scientific computing, C++ offered a more stable and production-ready foundation for building a scalable and performant ocean model from the ground up. Firedrake (Ham et al., 2023) is an example of a more specialized domain specific language. It provides a high-level interface with partial differential equations (PDEs) and underlying discretizations, but is not strongly supported on GPU architectures.

In designing Omega, we found the performance-portable library approach (option 4) to be the most promising. The existing C++ based libraries (Kokkos, Raja, YAKL) all offer similar capabilities, including data array abstractions for managing the CPU and GPU memory spaces, as well as a parallel\_for construct for kernel launches and parallel execution on the GPU. A number of additional utilities are also provided to support a performance-portable interface across the heterogeneous nodes. We began Omega with YAKL because it was the simplest and most light-weight library, specifically developed to port existing

90

95

Fortran atmosphere codes. However, due to the future risk associated with it being a single-developer package, we switched to the Kokkos library.

The choice of Kokkos required our development team to transition from Fortran to C++. This is a major change, as Fortran has been used by our group and the climate modeling community for many decades. We consider the model rewrite in C++ a worthwhile, long-term investment for its widespread support across all major HPC platforms. C++ benefits from decades of ecosystem development, robust support by compiler vendors, and a wealth of well-established libraries for MPI, parallel I/O, and performance portability frameworks like Kokkos.

Kokkos was already being used by the E3SM atmosphere component, and we had developed some internal project expertise with it. The E3SM Atmosphere Model in C++ (EAMxx) was designed from the ground up using C++ and Kokkos. EAMxx and its high-resolution counterpart, the Simple Cloud-Resolving E3SM Atmosphere Model (SCREAM), won the 2023 Gordon Bell Climate Prize for Modeling award for being the first global cloud-resolving model to run efficiently on an exascale supercomputer (Donahue et al., 2024). SCREAM was designed to provide sufficient parallelism to keep GPUs fully utilized, and surpassed one simulated year per compute day at global 3 km resolution.

A recent example of a Kokkos-based ocean model outside of DOE is LICOMK++ (Wei et al., 2024a). They showed performance portability across CPUs and HIP-based GPUs (Wei et al., 2024b). Like the Omega effort we describe here, this ocean model is still in the early stages of development, lacking some features found in more mature ocean models and relying on more uniform, regular meshes. Nonetheless, it will be a valuable point of comparison for Omega going forward. It should be noted that unlike the two efforts above, we were able to develop Omega with a small group composed primarily of domain scientists, without a dedicated computer science team. We have purposely written the code to be legible to domain scientists, simplifying some of the Kokkos abstractions for that purpose.

In this paper, we document the first phase of our future ocean model, Omega, using the Kokkos portability layer. The model is described in Section 2, including the governing equations, variable definitions, and discrete formulation. The code design in Section 3 explains the details of the model framework, Kokkos interface, and code organization. Section 4 describes four verification tests of increasing complexity. Section 5 provides Omega performance results on three architectures with comparisons on CPUs and GPUs, and with MPAS-Ocean. Conclusions are presented in Section 6.

# 2 Model Description

Omega Version 0.1.0 (Omega-V0) was created as a first version of the full Omega primitive equation model. It solves the shallow water equations (SWE), as well as the advection-diffusion equation for passive tracers. This is sufficient to test performance using the Kokkos library on CPUs and GPUs, as well as the framework functions described in Section 3.1. Omega-V0 has redundant vertical layers in order to test performance using arrays with a vertical index, but does not include any vertical advection or diffusion terms.

130

### 2.1 Governing equations

The shallow water equations govern the conservation of momentum and volume for an incompressible fluid on the rotating earth. Standard formulations may be found in textbooks on Geophysical Fluid Dynamics, such as those by Vallis (2017); Cushman-Roisin and Beckers (2011); Gill (1982); Pedlosky (1986). Here we follow the presentation in Bishnu et al. (2024), Section 2.1. In continuous form, the shallow water equations are

$$\frac{\partial \mathbf{u}}{\partial t} + (\mathbf{u} \cdot \nabla) \mathbf{u} + f \mathbf{k} \times \mathbf{u} = -g \nabla (h - b), \tag{1}$$

$$\frac{\partial h}{\partial t} + \nabla \cdot (h\boldsymbol{u}) = 0. \tag{2}$$

All variables introduced in this section are summarized in Table 1. Using a vector calculus identity, the non-linear advection term may be represented as

$$\boldsymbol{u} \cdot \nabla \boldsymbol{u} = (\nabla \times \boldsymbol{u}) \times \boldsymbol{u} + \nabla \frac{|\boldsymbol{u}|^2}{2}$$
(3)

$$= \{ \mathbf{k} \cdot (\nabla \times \mathbf{u}) \} \mathbf{k} \times \mathbf{u} + \nabla \frac{|\mathbf{u}|^2}{2}$$
 (4)

$$=\omega \boldsymbol{u}^{\perp} + \nabla K. \tag{5}$$

Thus the advection and Coriolis term may be combined together as

$$\boldsymbol{u} \cdot \nabla \boldsymbol{u} + f \boldsymbol{k} \times \boldsymbol{u} = (\omega + f) \, \boldsymbol{u}^{\perp} + \nabla K \tag{6}$$

$$=q\left(h\boldsymbol{u}^{\perp}\right)+\nabla K,\tag{7}$$

where q is the potential vorticity. This formulation, described in Section 2.1 of Ringler et al. (2010), is useful for the mimetic properties of potential vorticity and energy conservation in the TRiSK discretization (Thuburn et al., 2009).

The governing equations for Omega-V0 in continuous form are

$$\frac{\partial \boldsymbol{u}}{\partial t} + q\left(h\boldsymbol{u}^{\perp}\right) = -g\nabla(h-b) - \nabla K + \nu_2 \nabla^2 \boldsymbol{u} - \nu_4 \nabla^4 \boldsymbol{u} - C_D \frac{\boldsymbol{u}|\boldsymbol{u}|}{h} + \frac{\tau}{h}$$
(8)

$$\frac{\partial h}{\partial t} + \nabla \cdot (h\mathbf{u}) = 0 \tag{9}$$

$$\frac{\partial h\varphi}{\partial t} + \nabla \cdot (h\boldsymbol{u}\varphi) = \kappa_2 h \nabla^2 \varphi - \kappa_4 h \nabla^4 \varphi. \tag{10}$$

In order to bring these equations closer to the layered formulation of the upcoming full ocean model in Omega-V1, we have added Laplacian and biharmonic dissipation to the momentum equation, along with quadratic bottom drag and wind forcing. The thickness equation (9) is derived from conservation of mass for a fluid with constant density, which reduces to conservation of volume. The model domain uses fixed horizontal cells with horizontal areas that are constant in time, so the area drops out and only the layer thickness h remains as the prognostic variable. The tracer equation (10) is the conservation equation for a passive tracer (scalar), with only advective and diffusive terms. It is not included in the textbook shallow water equations, but is useful for us to test tracer advection in preparation for a primitive equation model in Omega-V1. In this equation set,

the tracer equation does not feed back into the momentum or thickness equations. It is written in a thickness-weighted form because the conserved quantity is the tracer mass. Here  $(h\varphi A)$ , where A is horizontal cell area, typically has units of tracer mass in kg, while  $\varphi$  has units of concentration in kg m<sup>-3</sup>. Since A is fixed, it is divided out, making (10) thickness-weighted, rather than volume-weighted. A derivation of the thickness-weighted tracer equation appears in Appendix A-2 of Ringler et al. (2013). The Omega-V0 governing equations do not include any vertical advection or diffusion. Although Omega-V0 includes a vertical index for performance testing and future expansion, vertical layers are currently redundant.

**Table 1.** Definition of variables

| symbol         | name                         | units                       | location | notes                                                          |
|----------------|------------------------------|-----------------------------|----------|----------------------------------------------------------------|
| b              | bottom depth                 | m                           | cell     | always positive                                                |
| $C_D$          | bottom drag                  | $\mathrm{m}^{-1}$           | constant | typically 0.001                                                |
| f              | Coriolis parameter           | $s^{-1}$                    | vertex   |                                                                |
| g              | gravitational acceleration   | $\rm m\;s^{-2}$             | constant |                                                                |
| h              | layer thickness              | m                           | cell     |                                                                |
| $oldsymbol{k}$ | vertical unit vector         | unitless                    | none     |                                                                |
| K              | kinetic energy               | $\rm m^2~s^{-2}$            | cell     | $K = \ \boldsymbol{u}\ ^2/2$                                   |
| q              | potential vorticity          | ${\rm m}^{-1}~{\rm s}^{-1}$ | vertex   | $q = (\omega + f)/h$                                           |
| t              | time                         | s                           | none     |                                                                |
| $oldsymbol{u}$ | velocity, vector form        | ${\rm m}~{\rm s}^{-1}$      | edge     |                                                                |
| $u_e$          | velocity, normal to edge     | $\rm m\;s^{-1}$             | edge     |                                                                |
| $u_e^\perp$    | velocity, tangential to edge | ${\rm m}~{\rm s}^{-1}$      | edge     |                                                                |
| $\kappa_2$     | tracer diffusion             | $\rm m^2~s^{-1}$            | cell     |                                                                |
| $\kappa_4$     | biharmonic tracer diffusion  | $\rm m^4~s^{-1}$            | cell     |                                                                |
| $ u_2$         | viscosity                    | $\rm m^2~s^{-1}$            | edge     |                                                                |
| $ u_4$         | biharmonic viscosity         | $\rm m^4\;s^{-1}$           | edge     |                                                                |
| arphi          | tracer                       | varies                      | cell     | units: $kg m^{-3}$ or similar                                  |
| au             | wind stress                  | Pa                          | edge     |                                                                |
| $\omega$       | relative vorticity           | $s^{-1}$                    | vertex   | $\omega = \boldsymbol{k} \cdot (\nabla \times \boldsymbol{u})$ |

# 2.2 Discretization

The horizontal domain is partitioned into polygonal finite-volume cells. Definitions of the mesh variables, differential operators and illustrative figures can be found in Ringler et al. (2010), Section 3, and are not reproduced here.

In discrete form, the governing equations are

$$\frac{\partial u_e}{\partial t} + \left[\frac{\mathbf{k} \cdot \nabla \times u_e + f_v}{[h_i]_v}\right]_e \left([h_i]_e u_e^{\perp}\right) = -g\nabla(h_i - b_i) - \nabla K_i + \nu_2 \nabla^2 u_e - \nu_4 \nabla^4 u_e - C_D \frac{u_e|u_e|}{[h_i]_e} + \frac{\tau_e}{[h_i]_e}$$

$$\tag{11}$$

$$\frac{\partial h_i}{\partial t} + \nabla \cdot \left( [h_i]_e u_e \right) = 0, \tag{12}$$

$$\frac{\partial h_i \varphi_i}{\partial t} + \nabla \cdot \left( u_e [h_i \varphi_i]_e \right) = \kappa_2 h_i \nabla^2 \varphi_i - \kappa_4 h_i \nabla^4 \varphi_i, \tag{13}$$

where subscripts i, e, and v indicate cell, edge, and vertex locations (i was chosen for cell because c and e look similar). Here square brackets  $[\cdot]_e$  and  $[\cdot]_v$  represent quantities that are interpolated to edge and vertex locations. The interpolation is typically centered, but may vary by method, particularly for advection schemes. For vector quantities,  $u_e$  denotes the normal component at the center of the edge, while  $u_e^{\perp}$  denotes the tangential component.

Documentation of operator convergence rates are provided in Bishnu et al. (2023) Section 4.1 and Figure 1. All TRiSK spatial operators demonstrate second-order convergence on a uniform hexagon grid, except for the curl on vertices, which is first order. The curl interpolated from vertices to cell centers regains second order convergence. The rates of convergence are typically less than second order on nonuniform meshes, including spherical meshes. Tracer advection uses center-weighted thickness and tracer values at each edge. The boundary conditions are no normal flow and no-slip. This is accomplished by setting the edge-normal velocity  $u_e = 0$  on the boundary for flux and vorticity calculations.

## 170 3 Code design

165

175

180

Omega-V0 has been designed to perform efficiently on modern parallel, hybrid HPC architectures. The design utilizes a domain decomposition of the unstructured mesh across parallel nodes with data communicated between the partitions using the MPI (2025,1993). Within a single shared-memory node, we have adopted the Kokkos (Trott et al., 2022a) programming model to map the computational work to either CPU cores (host) or GPU accelerators (device). This necessitated that Omega be written in the C++ programming language (Stroustrup, 1986, 2013). We have added some additional abstractions or aliases to simplify some of the Kokkos syntax and make it more accessible to Omega developers (see Section 3.2). Kokkos is a well-supported, portable framework (Trott et al., 2021) that has enabled us to create a performance-portable ocean model.

All components of Omega follow a process of design document writing and review, and then code writing, testing and review. Each feature is accompanied with a user's guide, developer's guide, and the original design document on the Omega documentation website (Asay-Davis et al., 2025b). This detailed information, created by each developer during code development, will serve as a comprehensive reference for the completed model.

#### 3.1 Framework

## 3.1.1 Domain Decomposition

As described above, the top level of parallelism is a domain decomposition of the horizontal mesh. We utilize the Metis library (Karypis, 2013) to perform the decomposition, given a mesh connectivity computed from JIGSAW (Engwirda, 2018). Unlike

the previous MPAS model, the decomposition is computed at startup with a call to the Metis library rather than being computed off-line. This eliminates a preprocessing step and the need to maintain partition files for different model configurations. The number of tasks is determined at run time from either the MPI environment when running a standalone ocean model or from a coupled model driver when Omega is run coupled within E3SM. The actual layout of MPI tasks across CPU cores and GPUs within a node can be set with job submission scripts. Multiple domain decompositions are supported so that some phases (e.g., a barotropic mode solver or analysis tasks) can be run in a different task configuration to optimize for communication or to enable a larger subgrid size.

## 3.1.2 Message Passing Infrastructure

Message passing is used to communicate data between the horizontal domains. The Omega base infrastructure layer provides simple interfaces for performing communications like the broadcast of data from a single task, the updating of domain halos and performing global reductions like sums across the global domain. All communication routines can determine whether the data exists on the host or device and can utilize GPU-aware MPI capabilities wherever available. The global sum function is bit-reproducible for all data types. For single-precision (32-bit) floating point types, the sums are performed in double precision (64-bit) and converted back to single precision. For double-precision floating point data, the sums are computed using the double-double algorithms of Knuth (1969) and Hida et al. (2008) following the implementation of He and Ding (2001).

We have implemented an MPI halo exchange module that handles the transfer of data across interfaces between adjacent partitions in a given domain decomposition. This implementation supports exchanges of multidimensional arrays of fundamental data types residing in either CPU or GPU memory. The module is designed to minimize latency by utilizing non-blocking MPI routines (i.e., MPI\_Isend and MPI\_Irecv) and supports a user-configurable halo width.

To maximize performance on GPU-accelerated systems, the halo exchange module can leverage GPU-aware MPI, enabled via a compile-time build flag. When built for GPU execution, halo elements are packed into and unpacked from contiguous buffers directly on the device using parallel kernels. With GPU-aware MPI enabled, send and receive buffers in device memory are passed directly to the MPI routines; otherwise, the packed send buffers must be copied from device to host for traditional host-staged MPI, and the received buffers are copied back from host to device before unpacking. Benchmarking on Frontier at Oak Ridge National Laboratory with GPU-aware Cray MPICH demonstrates that this approach significantly reduces halo exchange overhead, yielding approximately a 4–6× reduction in halo exchange time per time step compared to host-staged MPI at large node counts, where communication is latency dominated (Figure 1).

### 3.1.3 Other utilities

Configuration of Omega is done through an input configuration file in YAML (YAML, 2009) format. We use the yaml-cpp library (Beder, 2023) to read and parse the configuration on initialization. Logging of both informational and error messages are part of Omega's logging and error handling capabilities that are built on the spdlog library (Melman, 2023). This supports

235

**Figure 1.** Ratio of execution time per timestep for halo exchanges using host-staged MPI versus GPU-aware MPI on the Frontier supercomputer at Oak Ridge National Laboratory with Cray MPICH. Results are shown for three different planar mesh sizes, utilizing eight MPI tasks per node.

varying levels of error/log severity and messages can be written from either a master task or from all tasks, depending on a build-time configuration.

All input and output are performed in parallel using the SCORPIO library (Krishna et al., 2024) that writes distributed data using a runtime configuration of IO tasks. It supports both NetCDF (Unidata, 2023) and ADIOS (Godoy et al., 2020) formats. Multiple IO streams can be defined with each stream having its own frequency of input/output and its own set of fields. The details of each stream are specified by the user in the streams section of the input configuration file. Each field available for IO is defined within Omega using a field class that defines the metadata associated with the field and attaches/detaches the data array as needed. The field creation interfaces ensure that all required metadata are defined in accordance with the NetCDF CF metadata conventions (Eaton et al., 2024).

A time manager tracks model time in the context of a number of supported calendars. It uses integer arithmetic to avoid round-off in accumulated time. It is a reimplementation of the Earth System Modeling Framework (ESMF, 2020) time manager, that has been simplified for more clarity and streamlined by removing unnecessary functionalities, such as Fortran interfaces. It includes support for a model clock, time instants, time intervals (e.g., time step) and alarms for various model events like forcing and IO.

We use a profiling interface called Pacer to keep track of the computational time spent in various model processes using application level markers that designate beginning and end of each process. These timers are aggregated across multiple ranks and a summary report is generated when running in parallel. This timing infrastructure is based on our extensions to the General Purpose Timing Library (GPTL) (Rosinski, 2018).

250

255

### 3.2 Performance Portability with Kokkos

To achieve performance portability, Omega has adopted the Kokkos Programming Model (Trott et al., 2022b). The Kokkos Programming Model is implemented as a C++ library and provides abstractions necessary to achieve performance on the diverse set of modern computing architectures. Kokkos abstractions can be divided into abstractions for data storage (View, Memory Space, Memory Layout, and Memory Traits) and parallel execution (Execution Space, Execution Policy, and Execution Pattern). Omega builds its own abstractions on top of these fundamental components to provide a simpler interface for domain scientists.

For data storage, Omega uses the Kokkos View data structure. For convenience, type aliases are provided for commonly needed views of fundamental data types, such as

- Array1DI4: device-resident one-dimensional array of four-byte integers,
- Array3DReal: device-resident three-dimensional array of user-configurable floating-point type,
- HostArray2DI8: host-resident two-dimensional array of eight-byte integers,

and similarly for other combinations of ranks and types.

For parallel execution, Omega provides a parallelFor function, that can express parallel iteration over a multi-dimensional index range. Figure 2 shows how a simple Fortran loop nest is expressed in Omega. Internally, this function dispatches to

```
real, dimension(3, 4, 5) :: A
...
do k = 1, 5
do j = 1, 4
    do i = 1, 3
        A(i, j, k) = i + j + k
    end do
end do
end do
```

```
Array3DReal A("A", 3, 4, 5);

parallelFor({3, 4, 5},
    KOKKOS_LAMBDA (int i, int j, int k) {
        A(i, j, k) = i * j + k;
});
```

Figure 2. Multi-dimensional iteration expressed in Fortran (left) and using Omega abstractions (right).

the best performing (in the context of Omega) Kokkos execution policy for the chosen compute platform. Currently, we use Kokkos MDRangePolicy on CPU platforms, but opt to use a one-dimensional RangePolicy with manual index unpacking on GPUs, as this reduces GPU runtime overhead by replacing the more complex index mapping logic of MDRangePolicy with a simpler manual calculation performed within each thread. This simplification can lower instruction count and improve memory access patterns and cache utilization. Additionally, flattening the iteration space enables Kokkos's internal heuristics to more effectively select GPU kernel launch parameters, such as block size and grid configuration, thereby improving occupancy and

260

265

load balancing. On Frontier and Perlmutter GPU nodes, this approach yielded a 10–20% reduction in kernel execution time compared to MDRangePolicy.

Individual computations in Omega (for example, tendency terms or auxiliary variables) are implemented as C++ functors, which are classes that implement the function call operator. Functors can be called similarly to normal C++ functions, but may contain an internal state. In Omega, functors are used to represent computations for a given mesh element (e.g., vertex, cell, or edge) index and over a chunk of vertical levels. Our strategy is to design functors that perform computations over contiguous chunks of vertical indices with a chunk size known at compile time, to facilitate vectorization on CPUs. For GPU execution, the chunk size is set to 1 to distribute the workload across as many GPU threads as possible. To simplify the calling interfaces, Omega functors store as member variables the static data needs to implement their operation, such as mesh connectivity or geometry information. Variable input data are passed as arguments.

To give a concrete example, a functor that implements the kinetic energy gradient tendency term is shown in Figure 3. Its constructor takes a pointer to the HorzMesh object so that the functor can store pointers to the CellsOnEdge connectivity array and the DcEdge geometry array. The operator() implements the kinetic energy gradient computation for the edge index IEdge and over the range [KChunk \* VecLength, KChunk \* VecLength + VecLength) of vertical levels. This functor can then be used to compute the tendency term over the whole mesh by using the parallelFor function, as shown in Figure 4.

Omega tendencies are composed of multiple terms. The functor approach makes it possible to easily switch between computing multiple tendency terms in one parallel loop or in separate parallel loops. For example, given another functor that computes the ssh gradient term SSHGradOnEdge, the kinetic energy and the ssh gradients can be computed together or separately, as shown in Figure 5. Kernel fusion is a powerful optimization technique that often results in better performing code due to reduced overheads and data reuse. However, overuse of this optimization may result in high register usage, which can sometimes lead to worse performance. Therefore, having the flexibility to experiment with different splittings is important.

## 3.3 Code Organization and C++ Classes

Omega is organized into modularized classes to handle major pieces of the PDE solver such as Decomp, Halo, Mesh, State variables, Auxiliary variables, Timestepping, and Tendency terms. The decomposition of the mesh into local MPI rank subdomains is performed online in the Decomp class with the resulting local subdomain mesh represented in the Mesh class. The infrastructure necessary to perform message passing on the host and device between local subdomain halo regions is contained in the Halo class. The State class manages the prognostic variables, while the Auxiliary variable class stores and computes diagnostic quantities derived directly from the prognostic variables and used in the tendency terms, e.g., kinetic energy andpotential vorticity. The tendency terms are implemented as functors, where the operator method computes the tendency term for a single mesh location (e.g., cell or edge) and over a range of vertical levels. The constructor of each tendency functor takes in and stores static mesh information as private member variables, which simplifies the calling arguments in the PDE solution.

```
class KEGradOnEdge {
public:
   bool Enabled;
  /// constructor
   KEGradOnEdge(const HorzMesh *Mesh)
    : CellsOnEdge(Mesh->CellsOnEdge), DcEdge(Mesh->DcEdge) {}
  /// The functor takes edge index, vertical chunk index, and kinetic energy
   /// array as inputs, outputs the tendency array
   KOKKOS_FUNCTION void operator()(const Array2DReal &Tend, I4 IEdge, I4 KChunk,
                                   const Array2DReal &KECell) const {
      const I4 KStart
                           = KChunk * VecLength;
      const I4 JCell0
                           = CellsOnEdge(IEdge, 0);
      const I4 JCell1
                           = CellsOnEdge(IEdge, 1);
      const Real InvDcEdge = 1._Real / DcEdge(IEdge);
      for (int KVec = 0; KVec < VecLength; ++KVec) {</pre>
         const I4 K = KStart + KVec;
         Tend(IEdge, K) -= (KECell(JCell1, K) - KECell(JCell0, K)) * InvDcEdge;
      }
   }
private:
   Array2DI4 CellsOnEdge;
   Array1DReal DcEdge;
};
```

Figure 3. Kinetic energy gradient functor in Omega.

```
KEGradOnEdge KEGrad(Mesh);
parallelFor({NEdges, NChunks},
   KOKKOS_LAMBDA (int IEdge, int KChunk) {
   KEGrad(NormalVelocityTend, IEdge, KChunk, KECell);
});
```

**Figure 4.** Computation of kinetic energy gradient over the whole mesh.

```
KEGradOnEdge KEGrad(Mesh);
SSHGradOnEdge SSHGrad(Mesh);

parallelFor({NEdges, NChunks},
   KOKKOS_LAMBDA (int IEdge, int KChunk) {
   KEGrad(NormalVelocityTend, IEdge,
        KChunk, KECell);

SSHGrad(NormalVelocityTend, IEdge,
   KChunk, SSHCell);
});
```

Figure 5. Split (left) or fused (right) computation of two tendency terms.

# 3.4 Build and Internal Testing

The Omega build system, built on the widely adopted CMake (Kitware, Inc., 2023b) tool, establishes a robust framework for managing the compilation process. It operates in two distinct modes: standalone and E3SM component. In standalone mode, Omega generates a generic E3SM case and derives its build configurations from it. In contrast, the E3SM component build mode leverages build configurations provided by the CIME (Anderson et al., 2015) build system within an existing E3SM case. The build process, meticulously defined in the top-level CMakeLists.txt file, is segmented into four sequential steps: Setup, Update, Build, and Output.

A comprehensive testing strategy ensures Omega's quality assurance and continuous integration. All major Omega algorithms and software frameworks are rigorously validated using CTest (Kitware, Inc., 2023c), CMake's integrated testing tool. This enables the execution of functional tests, activated by setting OMEGA\_BUILD\_TEST=ON during the CMake configuration. These tests are critical to verify the correct functionality and integrity of the codebase.

Furthermore, nightly tests are developed and integrated with CDash (Kitware, Inc., 2023a) to maintain ongoing stability and performance. This integration facilitates automated reporting of test results, providing continuous feedback on the codebase's status. This robust testing infrastructure, which includes both CTest-based functional tests and CDash-driven nightly regressions, is paramount to ensuring the high quality and reliability of the Omega ocean model.

### 4 Verification Tests

A series of Omega-V0 tests were conducted to verify the accuracy of the model solution, and document the computing performance across several platforms. Convergence studies against exact solutions in idealized domains were conducted with

the manufactured solution, tracer transport, and barotropic gyre test cases. The global wind-driven Simulation was designed to introduce coastlines, bathymetry, and wind forcing, in order to test the workflow for realistic domains.

We developed a python package polaris to facilitate the set-up and execution of verification and validation tests for Omega. polaris is responsible for creating the MPAS mesh using the JIGSAW library (Engwirda, 2018), generating the initial condition, configuring the forward model run and linking the model executable, and conducting analysis including producing visualizations on the native MPAS mesh. polaris facilitates the creation of identical test cases for MPAS-Ocean and Omega, supporting the benchmarking of Omega implementations against MPAS-Ocean.

### 4.1 Manufactured Solution

The method of manufactured solutions is commonly used for the code verification of partial differential equations (PDE) solvers. Unlike code validation, which assesses whether a model captures the correct physics by comparing its results to experimental or observational data, code verification is a purely mathematical exercise that evaluates whether a code correctly implements the intended numerical method. The manufactured solution approach was formalized in the computational science literature by Salari and Knupp (2000) and further refined in Roache (2002). The key idea is to choose an exact solution, substitute it into the PDE, and include the residual terms as a source term. This enables the creation of analytic test cases for the full shallow water system, including non-linear terms. It stands out in this respect, as other shallow water test cases, such as the coastal Kelvin wave or the inertia-gravity wave test case (Bishnu et al., 2024) only provide analytic solutions to the linear, inviscid form of the equations. Therefore, the manufactured solution represents the single best test case for the verification of all terms in the model. We 'manufactured' our solution to match the test case described in detail in Bishnu et al. (2024) Section 2.10. However, as noted in that work, any smooth solution in space and time can be used, provided that the source terms are correctly defined. The test case verifies the time-stepping scheme along with the sea surface height gradient, Coriolis, and non-linear advection terms. We have only modified the source term to include both Laplacian and biharmonic dissipation.

The polaris system automates the testing of the manufactured solution for both MPAS-Ocean and Omega. The expected convergence rate is second order, as shown in Bishnu et al. (2024) Figures 13 and 19. These results are reproduced in Figure 6, which is generated by polaris using data from regular planar hexagonal meshes with grid cells of width 200, 100, 50, and 25 km. The corresponding time steps are 300, 150, 75, and 37.5 seconds, and the error was measured after 10 hours of simulation time. All tests use Laplacian and biharmonic viscosity coefficients of  $\nu_2 = 1.5e06 \text{ m}^2\text{s}^{-1}$  and  $\nu_4 = 5e13 \text{ m}^4\text{s}^{-1}$  respectively, classical fourth-order Runge-Kutta time-stepping (e.g. section 24.2 of Hamming (1973)), and a center-weighted thickness advection. The tracer equation (13) is not used in this test.

Individual operators such as the gradient, divergence, curl, and tangential velocities were verified in the early stages of model development. These used simple analytic functions such as sine waves on a doubly-periodic domain, where the exact solution was easily computed. The test setup follows Bishnu et al. (2023) Section 4.1, and was able to reproduce the second-order convergence for TRiSK operators shown in Figure 1 of that paper. The manufactured solution test is a superset of these tests, as it includes these individual terms.

**Figure 6.** Convergence plot for Omega with the Manufactured Solution Test, showing the L2 norm of the difference between the computed and analytic solution in sea surface height.

Figure 7. Sea Surface Height of the Manufactured Solution with a 25km grid.

# 340 4.2 Tracer transport on the sphere

Tracer transport was verified using a fixed angular velocity field and a tracer distribution that is advected around the sphere. This is named the cosine bell test case, and is available in polaris under cosine\_bell. It was first described in Williamson et al. (1992) but we use the variant from Sec. 3a of Skamarock and Gassmann (2011). A flow field representing solid-body rotation transports a bell-shaped perturbation in a tracer  $\psi$  once around the sphere, and the exact solution is the original distribution after one full rotation. The standard case evaluates error convergence with resolution, where the time step varies in proportion to the cell size. Another polaris test performs two runs of the cosine bell at coarse resolution, once with 12 and once with 24 cores, to verify the bit-for-bit identical results for tracer advection across different core counts. A final polaris test with the cosine bell configuration runs for two time steps at coarse resolution, then performs reruns of the second time step, as a restart run to verify the bit-for-bit restart capability for tracer advection.

Figure 8. Initial tracer concentration for the Cosine Bell Advection Test.

The cosine bell domain is an aquaplanet without continents, with a uniform depth of 300 m. The initial bell is defined by a passive tracer

$$\psi = \begin{cases} \left(\psi_0/2\right) \left[1 + \cos(\pi r/R)\right] & \text{if } r 

Figure 9. Convergence plot for Omega-V0 for the Cosine Bell Advection Test.

(Vallis, 2017, eqn. 14.43). Alternately, a similar barotropic gyre can be generated through a balance between wind stress and bottom stress, rather than viscosity. This variant is known as the Stommel Model (Stommel, 1948; Pal et al., 2023, App. B), which is not considered in this study.

The Munk Model serves as an excellent test case for the shallow water equations, as it is one of the few configurations with a physically meaningful circulation and an exact analytical solution. The wind stress field  $(\tau_x, \tau_y)$  is given by

$$\tau_x = \tau_0 \cos\left(\pi \frac{y}{L_y}\right),\tag{17a}$$

$$\tau_y = 0, (17b)$$

on a domain of width  $L_x \times L_y$ . We evaluate MPAS-Ocean and Omega against the analytic solution for the streamfunction  $\Psi$  under no slip boundary conditions (Vallis, 2017, p.743 equation 19.49):

380 
$$\Psi = \pi \sin(\pi y) \left( 1 - \tilde{x} - e^{-\tilde{x}/(2\epsilon)} \left[ \cos\left(\frac{\sqrt{3}\tilde{x}}{2\epsilon}\right) + \frac{1 - 2\epsilon}{\sqrt{3}} \sin\left(\frac{\sqrt{3}\tilde{x}}{2\epsilon}\right) \right] + \epsilon e^{(\tilde{x} - 1)/\epsilon} \right), \tag{18}$$

where  $\tilde{x} = x/L_x$  and  $\epsilon = L_m/L_y$ . Free slip boundary conditions are not available for either model and are not evaluated.

The wind-driven barotropic gyre is available in the polaris testing environment under the name barotropic\_gyre. It uses a dimensional version of the Munk Model in order to test Omega with realistic parameter values. The domain size is 1200 by 1200 km, with a resolution of 20 km. The maximum zonal wind stress amplitude,  $\tau_0 = 0.1$ ; the horizontal viscosity,  $\nu = 4e02 \text{ m}^2/\text{s}$ ; the Coriolis parameter,  $f = f_0 + \beta y$  with  $f_0 = 10^{-4} \text{ s}^{-1}$  and  $\beta = 10^{-10} \text{ s}^{-1}\text{m}^{-1}$ . Omega's advection term is linearized to permit a comparison against the analytical solution. The boundaries are non-periodic in both x and y, and the bottom topography is flat.

The case begins from rest with a uniform depth of 5000 m and zero sea surface height perturbations. It is spun up for three years, with a time step of 1 hour 23 minutes, chosen to satisfy the CFL condition with a Courant number of 0.25 and an assumed maximum velocity of 1 m s<sup>-1</sup>. Upon completion, the streamfunction is computed from the native edge-normal

Figure 10. Barotropic gyre test case after three years, showing the streamfunction for Omega (top) and MPAS-Ocean (bottom).

velocity. Both MPAS-Ocean and Omega have on the order of 10 percent differences in streamfunction magnitude from the analytic solution 10. After three years of simulation, small differences between the two models occur at the boundary. This may be due to different order of operations or compiler optimization.

## 4.4 Wind-driven Global Simulations

405

The final test of Omega adds realistic components to the configuration: Earth's coastlines and bathymetry on the sphere, climatological wind stress, bottom drag, and the full Coriolis parameter. This results in basin-wide circulations with western boundary currents such as the Gulf Stream and the Kuroshio current, and an Antarctic Circumpolar Current. This is the most realistic configuration one may attain with the shallow water equations, as variations in temperature and salinity, and the layered baroclinic dynamics are necessarily missing. Still, the wind-driven global simulation is an important step from the idealized box of the Munk Model, and demonstrates that the infrastructure for realistic geography and wind forcing is working properly. These components are essential for the upcoming layered version of Omega, where we can make quantitative comparisons to ocean observations.

There is no exact solution to the wind-driven global simulation. Therefore, we compare Omega-V0 to MPAS-Ocean, solving the shallow water equations with the same configuration. Both are run with the full nonlinear advection term, with a bottom drag coefficient of  $C=10^{-3}$ , Laplacian diffusion with  $\nu_2=10^3~\text{m}^2\text{s}^{-1}$  and biharmonic with  $\nu_4=1.2\times10^{11}~\text{m}^4\text{s}^{-1}$ . The coastal boundaries and realistic bathymetry for these single-layer simulations are interpolated from the GEBCO 2023 (GEBCO Bathymetric Compilation Group 2023, 2023) and BedMachine Antarctica v3 (Morlighem, 2022) which have been blended together between 60°S and 62°S. The ocean begins at rest with a uniform sea surface height of zero, and spins up for 40 days.

420

**Figure 11.** Detail of the icosahedral 7.5 km mesh, showing bottom depth in the Gulf of Mexico (top), and a close-up view of the Mississippi River Delta Region (bottom).

MPAS-Ocean and Omega read in the identical initial condition file, as they both use the MPAS mesh specification in a NetCDF file format.

A sequence of spherical icosahedral meshes were generated using the JIGSAW software via Compass (Asay-Davis et al., 2025a), the predecessor to polaris. The first mesh has 8 icosahedral subdivisions resulting in an average gridcell width of 30 km, and the width halves with each progressive subdivision. Here we show results for 10 subdivisions, with a resolution of 7.5 km and 7.44 million horizontal cells (Figure 11). A time step of 15 seconds is required at this resolution, which is similar to the barotropic time step in time-split layered ocean models, in order to satisfy the CFL condition for surface gravity waves. The wind forcing is constant in time, so there is no diurnal or seasonal variation. After a spin-up period of 40 days, one can observe the structure of the global circulation in the sea surface height (Figure 12), which is a proxy for the streamfunction. In the wind-driven shallow water system, strong currents develop along western boundaries and along deep sea ridges in the Southern Ocean (Figure 13). Omega and MPAS-Ocean produce the same circulation patterns, with differences of less than 5% throughout most of the domain. Visible differences along coastlines may stem from the accumulation of numerical errors in

**Figure 12.** Global wind-driven test case showing SSH in meters at day 40, with the 7.5 km icosahedral mesh. Results are for Omega (top), MPAS-Ocean (middle) and the difference (bottom).

Figure 13. Global wind-driven test case showing kinetic energy in  $m^2s^{-2}$  at day 40, with the 7.5 km icosahedral mesh. Results are for Omega (top), MPAS-Ocean (middle) and the difference (bottom).

430

435

450

energetic regions after 2.3e5 time steps. The 7.5 km mesh was run on 10 nodes, with a total of 1280 processors, on Perlmutter at the National Energy Research Scientific Computing Center (NERSC).

## 5 Computational Performance

Experiments were conducted to evaluate the computational performance of Omega-V0. The goals of this campaign are to measure: the computational throughput on both CPUs and GPUs; scaling with the number of compute nodes; and performance across a range of operational resolutions. The promise of the performance portability of Kokkos is tested in this section with three DOE platforms, which contain two types of CPUs and three different GPU designs. In order to take full advantage of DOE's computing resources, Omega must be able to achieve high throughput at large node counts with high resolution domains on all of these machines.

# 5.1 Hardware and Compiler Specifications

Performance testing was carried out on three of the largest supercomputers in the world: Frontier, Aurora, and Perlmutter. These were ranked second, third and twenty-fifth, respectively in the most recent Top500 list (Strohmaier et al., 2025), as shown in Table 2. Currently, the DOE owns the only three exascale computers on the list—El Capitan at 1.74 EFlop  $s^{-1}$ ; Frontier at 1.35 EFlop  $s^{-1}$ ; and Aurora at 1.01 EFlop  $s^{-1}$ , as measured by the High-Performance Linpack benchmark implementation. While we did not have access to El Capitan for this project, we were able to test Omega-V0's performance on architectures most relevant to DOE computing.

Frontier, Aurora, and Perlmutter provided a variety of chip designs to test the performance portability of the Kokkos library, as shown in Table 3. CPUs include AMD's EPYC 7763 and Intel's Xeon Max 9470. The three machines use three different GPU models: the AMD MI250X in Frontier; the Intel Data Center in Aurora; and the NVIDIA A100 Ampere in Perlmutter. Likewise, we tested three compilers on these machines: gnu on Frontier, intel on Aurora, and cray clang on Perlmutter (see Table 4).

# 5.2 Strong Scaling Tests

We conducted performance tests using the inertial-gravity wave shallow water test, available in the polaris suite under inertial\_gravity\_wave, and described in Section 2.6 of Bishnu et al. (2022). In order to mimic the performance requirements of a primitive equation ocean model, the Omega-V0 shallow water model was run with 96 identical vertical layers, five active tracers, the full non-linear advection terms, and the Laplacian and biharmonic terms active in both the momentum equation 11 and tracer equation 13. The choice of 96 layers was made as it is a multiple of 8, allowing for better vectorization. Wind forcing and bottom stress were not applied in these steps. The time-stepping scheme was chosen to be classical fourth-order Runge-Kutta.

The domain is doubly periodic on a Cartesian, regular hexagonal grid. Configurations of 1024x1024x96 and 2048x2048x96 gridcells are presented here. Performance times are equivalent for regular cartesian and unstructured spherical meshes, so the

| Top500 statistics          | Frontier                         | Aurora                          | Perlmutter                  |  |
|----------------------------|----------------------------------|---------------------------------|-----------------------------|--|
| Rank, June 2025            | 2                                | 3                               | 25                          |  |
| Linpack Performance (Rmax) | $1,353 \text{ PFlop s}^{-1}$     | $1{,}012~\mathrm{PFlop~s^{-1}}$ | $79 \text{ PFlop s}^{-1}$   |  |
| Theoretical Peak (Rpeak)   | $2,056 \text{ PFlop s}^{-1}$     | $1{,}980~\mathrm{PFlop~s^{-1}}$ | $113~\mathrm{PFlop~s^{-1}}$ |  |
| Nmax                       | 24,837,120                       | 28,773,888                      | 5,800,000                   |  |
| HPCG                       | $14{,}054~\mathrm{TFlop~s^{-1}}$ | $5,613 \text{ TFlop s}^{-1}$    | $1{,}905~\rm TFlop~s^{-1}$  |  |
| CPU cores for test         | 9,066,176                        | 9,264,128                       | 888,832                     |  |
| Power Consumption          | 24,607 kW                        | 38,698 kW                       | 2,945 kW                    |  |

**Table 2.** Performance statistics from the Top500 Supercomputers list, June 2025 (Strohmaier et al., 2025). Rmax is the maximum performance achieved using the LINPACK benchmark suite. Rpeak is the theoretical peak performance. Nmax refers to the size of the largest problem (specifically, the matrix size in a LINPACK benchmark) that a computer can solve. HPCG is the High-Performance Conjugate Gradient (HPCG) Benchmark results.

| Hardware          | Frontier                                         | Aurora                                                                          | Perlmutter                             |
|-------------------|--------------------------------------------------|---------------------------------------------------------------------------------|----------------------------------------|
| Manufacturer      | HPE                                              | Intel                                                                           | HPE                                    |
| Location          | Oak Ridge National Lab.                          | Argonne National Lab.                                                           | NERSC                                  |
| Installation Year | 2021                                             | 2023 (available 2025-Feb-14)                                                    | 2021                                   |
| Nodes             | 9856                                             | 10,624                                                                          | 4904 (1792 GPU; 3072 CPU; 40 login)    |
| CPU               | AMD Opt. 3rd Gen EPYC 2GHz                       | 2x Xeon Max 9470 2.4GHz                                                         | AMD EPYC 7763 2.45GHz                  |
| Cores per CPU     | 64                                               | 51                                                                              | 64 (GPU node); 128 (CPU node)          |
| GPU               | 4x AMD MI250Xs w/ 2 GCD                          | 6x Intel Data Center GPU Max                                                    | 4x NVIDIA A100 Ampere                  |
| CPU performance   | 2.51 TFlops/socket                               | 13.3 TFlops/socket                                                              | 2.51 TFlops/socket                     |
| GPU performance   | 47.9 TFlops (FP64)/GPU                           | 52.4 TFlops (FP64)/GPU                                                          | 9.7 TFlops (FP64)/GPU                  |
| Memory per node   | 512 GB of DDR4                                   | 1024 GB of DDR5, 128 GB HBM                                                     | 256 GB of DDR4 DRAM                    |
|                   | 64 GB HBM2E / GCD                                | 768 GB HBM for GPU                                                              |                                        |
| Memory bandwidth  | $204.8~\mathrm{GB}~\mathrm{s}^{-1}~\mathrm{CPU}$ | $2,870 \mathrm{Peak}\;\mathrm{GB}\;\mathrm{s}^{-1}\;\mathrm{CPU}\;\mathrm{HBM}$ | $204.8~\mathrm{GB~s^{-1}~CPU}$         |
|                   | $1600~\mathrm{GB}~\mathrm{s}^{-1}~\mathrm{GPU}$  | $19,\!660~\mathrm{Peak}~\mathrm{GB}~\mathrm{s}^{-1}~\mathrm{GPU}$               | $1555.2 \text{ GB s}^{-1} \text{ GPU}$ |
| Interconnect      | Slingshot-11                                     | Slingshot-11                                                                    | Slingshot-11                           |
|                   | Infinity Fabric                                  | PCIe 5.0 NIC-CPU connection                                                     | PCIe 4.0 NIC-CPU connection            |

**Table 3.** Hardware specifications for computers in this study, collected from Dongarra and Geist (2022); Oak Ridge National Laboratory (2025); Argonne National Laboratory (2025); NERSC (2025).

|       | Software for Omega-V0 tests                                               | Frontier                | Aurora                              | Perlmutter        |  |  |  |
|-------|---------------------------------------------------------------------------|-------------------------|-------------------------------------|-------------------|--|--|--|
|       | Operating System                                                          | HPE Cray OS             | SUSE Linux Enterprise Server 15 SP4 | HPE Cray OS       |  |  |  |
|       | Compiler                                                                  | gcc (SUSE Linux) 13.2.1 | Intel OneAPI DPC++ 2025.0.4         | Cray clang 18.0.1 |  |  |  |
|       | MPI                                                                       | cray-mpich/8.1.30       | mpich v5.0.0a1                      | cray-mpich/8.1.31 |  |  |  |
|       | Programming environment                                                   | PrgEnv-gnu/8.5.0        | oneapi/release/2025.0.5             | PrgEnv-cray/8.6.0 |  |  |  |
| Table | <b>Table 4.</b> Software for performance tests presented in this section. |                         |                                     |                   |  |  |  |

former is used here for convenience. Its horizontal grid cell count can easily be incremented by factors of two to produce a sequence of grid resolutions. The number of horizontal gridcells is approximately one million for the 1024x1024x96 domain and four million for the 2048x2048x96 domain. This compares to recent publications of 235 thousand horizontal cells by 64 vertical layers for the low-resolution global MPAS-Ocean E3SM domain (Smith et al., 2025), and 3.7 million by 80 vertical layers for the high-resolution 6 to 18 km MPAS-Ocean domain (Caldwell et al., 2019). Omega-V1 will be a full ocean model with additional computations such as vertical advection and mixing, equation of state, pressure computation, and physics parameterizations. Despite this, the current shallow water configurations provide a good preliminary representation of the performance comparison between Omega and MPAS-Ocean, between CPU and GPUs, and scaling to large node counts. For these tests, MPAS-Ocean has some of its primitive equation terms disabled so that it is solving the identical equations as Omega-V0. MPAS-Ocean was not tested on Aurora, the newest machine, because the purpose of three machines was to demonstrate the versatility of Omega on different hardware, and Frontier and Perlmutter were considered sufficient for the Omega versus MPAS-Ocean comparison.

Performance results for Omega-V0 are shown in Figure 14 for the 1024x1024x96 mesh, and in Figure 15 for the 2048x2048x96 mesh. Corresponding results for MPAS-Ocean are shown in Figures 16 and 17. In all cases, computation (blue lines) scales better than halo communication (green line), which is expected (Bishnu et al., 2023). Inter-node communication can be highly variable, depending on the competing traffic on the interconnect. Each point on these plots represents the time per timestep averaged over 5 simulations of 12 timesteps each, excluding start-up and I/O time. Since communication does not scale well with increasing node counts, low resolution configurations exhibit poor scaling due to insufficient computational intensity. This effect is more pronounced on GPUs (right column) than on CPUs (left column). In the "GPU" simulations, all CPUs and GPUs on each node are fully utilized for the timing test. As expected, the problem of poor scaling at a particular node count can be alleviated by running the model with higher resolution.

**Figure 14.** Strong scaling of Omega-V0 for the 1024x1024x96 resolution on Frontier (top), Aurora (middle) and Perlmutter (bottom), showing CPU-only simulations (left column), and GPUs with CPUs (right column). The colors separate the total (red) between the inter-node halo communication (green) and the on-node computation (blue). Start-up time and I/O are not included.

Figure 15. Same as Figure 14 but for the 2048x2048x96 resolution.

**Figure 16.** Strong scaling of MPAS-Ocean for the 1024x1024x96 resolution on Frontier (top) and Perlmutter (bottom), showing CPU-only simulations (left column), and GPUs with CPUs (right column).

**Figure 17.** Same as Figure 16 but for the 2048x2048x96 resolution.

|                                    |            | Frontier | Frontier | Aurora     | Aurora  | Perlmutter | Perlmutter |
|------------------------------------|------------|----------|----------|------------|---------|------------|------------|
|                                    |            | CPU      | GPU+CPU  | CPU        | GPU+CPU | CPU        | GPU+CPU    |
| WC time/timestep s, full node      | Omega      | 8.5      | 0.18     | 4.3        | 0.18    | 8.5        | 1.4        |
|                                    | MPAS-Ocean | 12.0     | 8.5      | na         | na      | 12.0       | 4.4        |
| number of CPUs/node used for test  |            | 56       | 32       | 104        | 6       | 64         | 64         |
| number of GPUs                     |            | 0        | 4        | 0          | 6       | 0          | 4          |
| power, TDP watts per CPU or GPU    |            | 280      | 560      | 350/socket | 600     | 280        | 300        |
| power, TDP watts per node          |            | 280      | 2520     | 700        | 4300    | 280        | 1480       |
| Throughput (model step/WC time)    | Omega      | 0.118    | 5.556    | 0.233      | 5.556   | 0.118      | 0.714      |
|                                    | MPAS-Ocean | 0.083    | 0.118    | na         | na      | 0.083      | 0.227      |
| Throughput per watt for 1000 steps | Omega      | 0.420    | 2.205    | 0.332      | 1.292   | 0.420      | 0.483      |
|                                    | MPAS-Ocean | 0.298    | 0.047    | na         | na      | 0.298      | 0.154      |
| Throughput per watt                | Omega      | 1.00     | 5.25     | 1.00       | 3.89    | 1.00       | 1.15       |
| relative to Omega CPU              | MPAS-Ocean | 0.71     | 0.11     | na         | na      | 0.71       | 0.37       |

Table 5. Timing for the 2048x2048x96 on four nodes, with thermal design power and throughput per watt power consumption.

Next, we compare throughput on CPU-only nodes versus when GPUs are added, and between Omega and MPAS-Ocean. To do this, we fix our comparison at four nodes, all within the "perfect scaling" regime, using the 2048x2048x96 mesh. These comparisons are not sensitive to the choice of resolution, as for each case, the 2048x2048x96 timing is almost exactly four times that of 1024x1024x96, demonstrating ideal weak scaling. The average wallclock time per time step is provided in the first two rows of Table 5. There are several ways to measure the speed-up when transitioning from CPU-only nodes to nodes with both CPUs and GPUs. The simplest method is to take the ratio of the compute times when the full resources of each node are utilized. For Omega-V0, this yields speed-ups of 47x on Frontier, 22x on Aurora, and 6.1x on Perlmutter, as shown in the top row of each arrow on Figure 18. Another comparison involves using the full CPU set versus a single GPU, which results in speed-ups of 12x, 3.7x, and 1.5x for Omega-V0 on these machines. However, one could argue that modern supercomputers are designed to deliver high GPU throughput, and the CPUs are simply helpers to coordinate the GPU computations. A major hardware design consideration is the reduced power usage per flop for GPUs, as the full supercomputer must aim to maximize computational throughput while minimizing total power consumption. To this end, we estimate the computational efficiency of our models with the thermal design power (TDP) of each chip (row 4 of Table 5). For example, on Frontier, the AMD EPYC 2GHz CPU is rated at 2.5 TFLOPs of double-precision performance and 225-280W TDP (HPCwire, 2021). In contrast, Frontier's AMD MI250X GPU specifications state 47.9 TFLOPs for 500-560W TDP (AMD, 2025), for a total of 191.6 TFLOPs and 2000-2240W TDP for the four GPUs on a single node. This means that the lion's share of computing and power consumption on Frontier takes place on the GPUs. Thus, the most meaningful comparison of code performance between CPUs and GPUs for a new model is based on the computational throughput per watt of power consumption. For Omega-V0,

**Figure 18.** A diagram of the speed-up factors when including the GPUs on each machine, for Omega-V0 and MPAS-Ocean. Times are based on four-node results on the 2048x2048x96 resolution shown in Table 5. The speed-up per watt uses the thermal design power of each CPU and GPU.

this metric shows performance improvements of 5.3x on Frontier, 3.6x on Aurora, and 1.2x on Perlmutter. Using this same method, these numbers for MPAS-Ocean are 0.16x on Frontier and 0.5x on Perlmutter, indicating a reduction in computational throughput per watt. Omega's relative performance is further highlighted in head-to-head comparisons on each chip: Omega-V0 is 1.4x faster than MPAS-Ocean on the AMD EPYC CPU, 3.1x faster with Perlmutter's NVIDIA A100 Ampere GPU, and 47x faster on Frontier's AMD MI250Xs. These results underscore the effectiveness of Omega's performance-portable design based on C++ and the Kokkos library.

## 6 Conclusions

This paper documents the governing equations, design philosophy, coding implementation, verification, and performance of Version 0.1.0 of the Ocean Model for E3SM Global Applications (Omega-V0). Version 0 is the first step towards a layered non-Boussinesq ocean model that can be used for realistic global applications as a component within E3SM. The motivation for rewriting the ocean model is to create a code base that is resilient to changing supercomputer architectures. We found

that our previous framework of Fortran code with MPI, OpenMP, and more recently OpenACC was not suitable for the new exascale computing landscape within DOE.

The key to the new Omega design is *performance portability*. The investment into developing a code base from scratch will pay off as new architectures are introduced, because the underlying Kokkos library will be updated and optimized for new machines while the Omega code may remain unchanged. Moving from Fortran to C++ offers the additional advantages of more standard libraries, modern code abstractions, and a language familiar to the next generation of developers.

The verification of Omega-V0 included convergence against exact solutions for the nonlinear shallow water equations using a manufactured solution test case, and for tracer advection on the sphere using a cosine bell test case. The barotropic gyre test case adds wind forcing, solid boundaries, and viscosity in an idealized domain, while the wind-driven global simulations validate our workflow with coastlines and bathymetry on a rotating earth. These tests are all automated and available in our polaris package, including the generation of initial conditions, statistical analysis, and visualization. Comparisons with exact solutions and MPAS-Ocean simulations provide confidence that Omega-V0 is working as expected.

Performance results on GPUs are of particular importance for this study, as that is the driving purpose of Omega. Omega-V0 is significantly faster on GPUs than on CPUs, as measured on a per node, per GPU, or per watt basis. Performance measurements on Frontier and Aurora, two of the world's fastest exascale computers, were quite promising. The speed-up from full-node CPU-only to full-node with GPUs was 47x on Frontier, 22x on Aurora, and 6.1x on Perlmutter (this is 12x, 3.7x, and 1.5x, respectively, on a per-GPU basis). Regarding energy consumption, improvement in throughput from CPUs to GPUs on a per Watt basis was 5.3x on Frontier, 3.6x on Aurora, and 1.2x on Perlmutter. This means that Omega's central design principle of performance portability was demonstrated on the exascale architectures that are most relevant to the DOE. In addition, performance tests were conducted to 128 nodes with high-resolution domains of 4 million horizontal cells and 96 layers. Compute times scale nearly perfectly to 128 CPU nodes and 32 GPU nodes. Good scaling to more nodes can be achieved with higher resolution configurations. Direct GPU-to-GPU communication was an important factor for successful Omega-V0 simulations on GPUs.

MPAS-Ocean is an important standard of comparison because it is the current ocean model in E3SM, and Omega is the candidate replacement. Omega-V0 is 1.4 times faster than MPAS-Ocean on CPUs. They use the same mesh specification, array structure, and indirect addressing of horizontal neighbors. Thus, the performance gains on CPUs can be attributed to improved optimization and memory layout in C++ and Kokkos. The speed-ups of MPAS-Ocean to Omega is particularly notable on GPUs, with a 4.7x speedup on Frontier and a 3.1x speedup on Perlmutter. In these tests, Omega-V0 and MPAS-Ocean had identical configurations and computed the same shallow water terms. The performance results confirm that MPAS-Ocean was constrained by the OpenACC approach on GPUs, whereas Omega has the potential to deliver faster simulations on GPU-based exascale computers.

Omega Version 1 will be a layered non-Boussinesq ocean model intended for real-world simulations. The underlying Kokkos framework will remain the same, but with additional terms for vertical advection and diffusion, an equation of state, hydrostatic pressure, and higher-order tracer advection. Version 1 will have similar capabilities as MPAS-Ocean in Ringler et al. (2013), and will be compared to realistic climatology. Version 2 will add coupling capability for surface fluxes within E3SM as in

550

Petersen et al. (2019), and more advanced parameterizations. The improved performance of Omega on GPUs, along with the atmospheric component EAMxx (Donahue et al., 2024), will allow E3SM to pursue state-of-the-art science on the world's newest and largest exascale supercomputers.

Code and data availability. Omega Version 0.1.0 is available at https://zenodo.org/records/17418901 and in the E3SM code repository under the tag Omega-v0.1.0-alpha.1 (https://github.com/E3SM-Project/Omega/releases/tag/Omega-v0.1.0-alpha.1) This version has the DOI 10.11578/dc.20250723.1 and is documented by DOE CODE at https://doi.org/10.11578/dc.20250723.1 (Petersen et al., 2025). Within the E3SM repository, Omega may be compiled as a standalone application by running cmake in the components/omega subdirectory. The Omega User's Guide may be found at https://docs.e3sm.org/Omega/omega. The testing framework is polaris version 0.7.0 (https://doi.org/10.5281/zenodo.15470123).

Author contributions. Code development, testing, and timing were conducted by all authors. Omega framework development was led by PJ, with team members SRB, YK, BO, MW. Shallow water model code developers included SRB, HK, AM, BO, MW. Testing, including the polaris development, was led by XSAD and CB with contributions by SB, SRB, MP, AB, KS. Performance measurement and improvements on three DOE computers were by MP, YK, AM, KR, SS, MW. Project management was by LVR, MP, SRB. The manuscript writing was led by MP, with contributions by all authors.

Competing interests. The authors declare no competing interests

Acknowledgements. Omega development is supported by the Energy Exascale Earth System Model (E3SM) project funded by the U.S. Department of Energy (DOE) Office of Science, Office of Biological and Environmental Research (BER). KR was additionally supported by the DOE's Los Alamos National Laboratory (LANL) LDRD Program and the Center for Nonlinear Studies.

This research used computational resources provided by: the National Energy Research Scientific Computing Center (NERSC), a DOE Office of Science User Facility supported by the Office of Science of the DOE under Contract No. DE-AC02-05CH11231; Oak Ridge Leadership Computing Facility at the Oak Ridge National Laboratory, which is supported by the Office of Science of the U.S. DOE under Contract No. DE-AC05-00OR22725; Argonne Leadership Computing Facility, a U.S. DOE Office of Science user facility at Argonne National Laboratory and is based on research supported by the U.S. DOE Office of Science-Advanced Scientific Computing Research Program, under Contract No. DE-AC02-06CH11357.

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
