# Peer review of "The Ocean Model for E3SM Global Applications: Omega Version 0.1.0. A New High-Performance Computing Code for Exascale Architectures"

_EGUsphere, 2025_

## Author Comment (AC1)

The authors thank the reviewers for the helpful comments, which improved the content and readability of the paper.

**Reviewer #1**

**General Comments**

This manuscript describes Omega-V0.1.0, a new C++/Kokkos-based ocean model for E3SM targeting performance portability across heterogeneous CPU/GPU architectures. The paper provides a clear scientific motivation for the rewrite from MPAS-Ocean, presents the governing equations and discretization in sufficient detail, and includes a broad set of verification tests and multi-platform performance benchmarks. The performance results, especially on multiple exascale-class GPU systems, are a valuable contribution to the community and align well with the objectives of GMD model description papers.

That said, several clarifications are still needed to strengthen reproducibility and to help readers interpret key results. In particular, the paper should provide more concrete explanations of why OpenACC offloading was limited in MPAS-Ocean, supply missing experimental details for the benchmarks, and expand the discussion of some performance claims (for example, regular versus unstructured mesh equivalence, CPU–GPU work partitioning). I also encourage the authors to discuss how the current performance conclusions are expected to extend to Omega-V1 when more complex physical parameterizations are added.

Overall, the manuscript is strong and suitable for publication after minor-to-moderate revisions focused on clarification and consistency.

**Specific Comments**

1. **Programming model taxonomy (around line 41):**

   The authors list four competing GPU programming approaches. Given the focus on portability, it would be useful to briefly mention recent language-standard based parallel models (for example, C++ and Fortran standard parallelism), and position them relative to the four categories already listed.

   **Author response:** Changes to language standards are in progress to express loop-level parallelism and accelerator models, but are not yet implemented widely and will take some time before they can be considered as a portable option. We have added a statement to that effect.

2. **Limitations of OpenACC in MPAS-Ocean (around line 56):**

   The manuscript explains that only about half of MPAS-Ocean could be accelerated with OpenACC and that this led to small kernels and poor throughput. Please add a concise, concrete explanation of which specific structural aspects of MPAS-Ocean prevented directive-based offload (for example, dynamic data structures, or control-flow

complexity).

**Author response:** We were trying to keep this initial justification section brief, but have added text that mentions the use of community based libraries that could not be accelerated, the MPI-dominated barotropic mode and the complexity of the tracer data structure and related loop complexity.

3. **Engagement of domain scientists and transition strategy (around line 106):**

The text states that Omega was developed by a small group mainly composed of domain scientists, and that Kokkos abstractions were simplified for legibility. Given that Omega-V1/V2 will require substantial physics and infrastructure development, it would be valuable to comment on how the Omega developer community is expected to grow (e.g., anticipated contributors from E3SM and the broader ocean/atmosphere community) and on practical strategies for enabling uptake by scientists less familiar with C++.

**Author response:** Our primary strategy is providing extensive documentation, with templates and examples, for domain scientists with less experience in C++ and Kokkos. Text has been added to that paragraph to address this point.

4. **Halo-exchange benchmark reproducibility (Figure 1 discussion around line 210):**

The description of GPU-aware MPI and the observed 4–6× speedup is clear, but key experimental parameters are missing. Please specify halo width, number and type of variables communicated per step, whether variables were packed separately or aggregated, and total and per-call message sizes.

**Author response:** In Omega-V0, three types of halo exchanges are performed during each time step, each with different message sizes: layer thickness, normal velocity, and an aggregated set of five tracer fields. All tests reported here use a halo width of three cells. Text has been added to explicitly state these details. These simulations use a fourth-order Runge-Kutta time stepper, which performs each type of exchange twice per time step (i.e., six total exchanges). For each exchange and for each communicating MPI task, data are packed into a contiguous buffer and communicated via a separate MPI call; variables are not aggregated across exchange types. In the single-node configuration, each MPI task communicates with five other MPI tasks, while in all other configurations, the average MPI task communicates with six MPI tasks. A new plot has been added that reports the average data sent per MPI task per time step, aggregated over all communicating tasks and all six halo exchanges. Across all configurations, approximately 11% of the communicated volume corresponds to the layer thickness exchanges, 34% to the normal velocity exchanges, and 55% to the tracer exchanges.

5. **Contiguous vertical chunk strategy and layout choice (around line 262):**

   Please clarify whether you tested or considered other memory orderings of the 3-D fields in Kokkos (changing which index is contiguous), and why the current choice (vertical index contiguous) is expected to be optimal across CPU and GPU architectures. In particular, for vertically dependent physics, non-coalesced access on GPUs could become a bottleneck; a short justification or discussion of tested layouts would be helpful.

   **Author response:** Only the vertically contiguous memory layout has been tested. This is the same memory layout that MPAS-Ocean uses. The EAMxx atmosphere model also uses this layout on both CPUs and GPUs, and is able to obtain good performance. Which layout is optimal for a given architecture is a difficult question to answer, since it depends on the choice of numerical algorithms. However, numerical algorithms are also tailored to the layout, making it hard to make a fair comparison between different memory layouts. To give a concrete example, future versions of Omega will need to solve a tridiagonal system for each column using a batched tridiagonal solver. On GPUs the classical Thomas algorithm parallelized over columns with the Omega memory layout leads to non-coalesced memory accesses. However, using the parallel cyclic reduction algorithm in shared memory is efficient with this layout, and exposes more parallelism in the vertical. In general, we believe that as long as most computations that have vertical dependencies can be parallelized, the vertically contiguous memory layout might be optimal. Kokkos provides efficient implementations of batched reduce and scan operations that can be used to express many such computations. Only for a select few, we might need to write different implementations for CPUs and GPUs. A paragraph summarizing this answer has been added at the end of Section 3.2.

6. **Regular Cartesian vs. unstructured spherical mesh performance (line 450):**

   The claim that performance is "equivalent" between regular Cartesian and unstructured spherical meshes is not explained. Please clarify what metric "equivalent" refers to and why indirect or irregular accesses in unstructured meshes do not measurably degrade performance.

   **Author response:** The regular hexagon cartesian mesh is treated as an unstructured mesh. There are no special data structures that take advantage of the physical regularity of the hexagon grid. That choice was made because the target application is unstructured spherical meshes, and the regular hex grids are just for performance testing. The performance is equivalent because both cartesian and spherical use the same indirect-addressed data structures. Text has been added to state this.

7. **Grid size notation and horizontal cell counts (line 450):**

The mesh is described as a regular hexagonal grid, and the test cases are labeled as 1024×1024×96 and 2048×2048×96. However, the mapping between the "1024×1024" notation and the reported horizontal cell counts (approximately one million and four million, respectively) is not obvious for a hexagonal mesh. Please add a brief explanation of what the 1024 and 2048 represent and how these translate to the stated horizontal cell numbers.

**Author response:** Thanks. We added text to this section to be more specific.

8. **CPU–GPU work partitioning in GPU runs (around line 470):**

The manuscript notes full utilization of CPUs and GPUs. Please describe how workload sharing between CPU and GPU is determined: automatic or manually tuned.

**Author response:** In the current implementation, the workload is not actually shared between CPUs and GPUs during GPU builds. The vast majority of computational work is executed on GPUs, while CPUs are primarily used for tasks like flow control, kernel launches, synchronization, and I/O. This sentence might be responsible for the confusion: "Within a single shared-memory node, we have adopted the Kokkos programming model to map the computational work to either CPU cores (host) or GPU accelerators (device)." This statement references the capability within Kokkos to map work to either host or device, but in Omega-V0 this is used exclusively to target the device during GPU builds. Future versions may utilize CPUs more to share the workload. A clarification in the manuscript to the above sentence has been added to make explicit that the compute-intensive work is handled entirely by the GPUs.

9. **Different CPU counts in Table 5 GPU vs. CPU-only runs:**

Table 5 uses fewer CPUs in GPU simulations than in CPU-only simulations. Please explain why the CPU count differs.

**Author response:** The difference reflects differences in the optimal execution models for the various cases. In Omega GPU builds, the compute-intensive work is handled by the GPUs, and the current implementation maps one MPI task to one CPU and one GPU. An optimal CPU-to-GPU ratio is generally 1, since using more than one CPU per GPU would require CPUs to share a GPU, leading to contention for kernel execution; conversely, using fewer CPUs reduces the number of mesh partitions and allows larger GPU kernels, improving computational efficiency relative to kernel-launch overhead. In CPU-only builds of Omega, these constraints do not apply, and all CPU cores on each node are used to maximize performance. In these tests, Table 5 reports throughput per watt comparisons between Omega and MPAS-Ocean, and both models use the same CPU counts within each configuration to ensure a fair comparison. At present, not all computational work in MPAS-Ocean has been ported to GPUs, and significant portions rely on OpenMP threading on CPUs, which motivates using higher CPU counts per node

in the GPU-enabled configurations on Frontier and Perlmutter. On Aurora, MPAS-Ocean was not tested; therefore, an optimal configuration of one CPU per GPU is used for Omega. The manuscript has been revised to include this motivation.

10. **Figure 7 missing reference:**

Figure 7 is not cited in the text. Please either reference and explain it or remove it.

**Author response:** Reference added to previous paragraph. Thank you.

11. **Convergence comparison with and without FCT (around line 362):**

Omega's tracer transport tests are conducted without FCT, whereas the manuscript reports the MPAS-Ocean convergence rate only for the FCT case (2.42). To enable a clearer like-for-like comparison, please also provide the MPAS-Ocean convergence rate *without* FCT and discuss whether that baseline is comparable to Omega's 1.36 rate.

**Author response:** We tested MPAS-Ocean with the identical mesh files and reduced order advection scheme, and obtained the same convergence rate. This was noted in the text.

12. **CPU runtime identity across machines (Figure 18):**

CPU runtimes on Frontier and Perlmutter are identical despite different compilers being used. Please double-check and add a brief comment confirming correctness if intended.

**Author response:** We retested and updated the values, which are within a few percent but not identical. You can also see this by comparing CPU Frontier to CPU Perlmutter in Figures 14-17. We expect them to be close but not identical. We added a sentence to point out that the compilers differ.

13. **Why Omega outperforms MPAS-Ocean on GPUs (around line 527):**

The reported GPU speedups of Omega over MPAS-Ocean are very large. However, the benchmark configuration targets a relatively simple shallow-water system with passive tracers and does not include the more complex, branching-heavy physical parameterizations that often challenge directive-based approaches. For such a comparatively regular workload, one might expect OpenACC to achieve reasonably high GPU efficiency as well. It is therefore unclear why the performance gap remains so dramatic. Please expand the discussion to identify which kernels or design choices dominate the difference (e.g., memory layout, kernel fusion/granularity, indirect addressing, communication overlap, or data movement), and explain concretely why OpenACC fails to reach similar efficiency for this specific configuration.

**Author response:** The MPAS-O OpenACC port was a very basic port with limited optimization. In particular, because it was only partially ported, we needed to retain host/device copies of the model state and the MPAS code structure prevented any kernel fusion. The MPAS MPI infrastructure was not GPU-aware so all communications were host-based. All of this required more data movement between host and device and limited any ability to reduce kernel launch overhead. Omega includes many of these optimizations. We have added some text to describe this.

14. **Absolute performance metrics:**

The performance analysis is currently presented almost entirely in terms of *relative* comparisons (across machines and against MPAS-Ocean). While these are useful, the absence of *absolute* performance metrics makes it difficult to assess efficiency against hardware limits or to compare with other studies. Please add at least one absolute metric (e.g., achieved memory bandwidth/FLOPS, or fraction of peak) to complement the relative results and strengthen the performance section.

**Author response:** Absolute performance metrics have been added to the new section 5.3. This includes the new Figures 19 and 20, and Table 6.

15. **Performance outlook when physical parameterizations are added:**

Omega-V0 benchmarks a relatively regular, shallow-water workload with passive tracers. Omega-V1 is expected to include more complex processes such as vertical advection and mixing, equation of state, pressure computation, and physics parameterizations. These additions often introduce more branching, irregular memory access, and heterogeneous kernel costs than the current configuration. Please include a short discussion on how the present performance conclusions are expected to translate to Omega-V1. For example:

· which future modules are anticipated to be performance-critical or memory-bound,

· whether the current kernel design strategy (functor granularity, vertical chunking, policy choices) is expected to remain optimal, and

· how OpenACC vs. Kokkos performance might change once less regular physics kernels are introduced.

Even a qualitative outlook would help readers assess the generality of the current performance results.

**Author response:** Thank you for the suggestion. A paragraph on the future performance outlook was added to the end of section 5.3.

**Technical Corrections**

· Replace nonstandard capitalization "Nvidia" → "NVIDIA" consistently.
**Author response:** Done.

· Define abbreviations at first use (SSH).
**Author response:** Done.

· Fix minor typos:

o "kinetic energy andpotential" → "kinetic and potential energy" (line 285)

o "analytic solution 10." → "analytic solution (Figure 10)" (line 392)

o "Simulation" → "simulation" (line 307).
**Author response:** Done.

· Replace e-notation in prose (for example, "1.5e06") with "1.5 × 10$^6$".
**Author response:** Done.

· The sentence starting "The tendency terms are …" (around line 286) largely repeats the content of the preceding paragraph (around lines 259–). Please remove it or merge the two passages to avoid redundancy.
**Author response:** removed.

· Figure 2 aims to show equivalent multi-dimensional iteration in Fortran (left) and Omega abstractions (right), but the right panel uses A(i,j,k)=i*j+k whereas the left uses i+j+k. This appears to be a typo and should be made consistent.
**Author response:** Done.

**Overall recommendation:** Minor revision. The required changes are mainly clarification for reproducibility and a small set of consistency and formatting fixes, with an added request to outline how performance expectations extend to Omega-V1 physics.